# Health Benefits of Residence at Moderate Altitude Do Not Reduce COVID-19 Mortality

**DOI:** 10.3390/ijerph192316074

**Published:** 2022-12-01

**Authors:** Johannes Burtscher, Gregoire P. Millet, Barbara Leitner, Martin Burtscher

**Affiliations:** 1Department of Biomedical Sciences, University of Lausanne, CH-1005 Lausanne, Switzerland; 2Institute of Sport Sciences, University of Lausanne, CH-1015 Lausanne, Switzerland; 3Directorate Population, Statistics Austria, A-1110 Vienna, Austria; 4Department of Sport Science, University of Innsbruck, A-6020 Innsbruck, Austria; 5Austrian Society for Alpine and High-Altitude Medicine, A-6020 Innsbruck, Austria

**Keywords:** COVID, coronavirus, altitude, hypoxia, mortality, aging

## Abstract

Moderate altitude (1000–2000 m above sea level) residence is emerging as a protective factor from the mortality of various causes, including of cardiovascular diseases. Conversely, mortality from certain respiratory diseases is higher at these altitudes than in lowlands. These divergent outcomes could indicate either beneficial or detrimental effects of altitude on the mortality of COVID-19 that primarily infects the respiratory tract but results in multi-organ damage. Previous epidemiological data indeed suggest divergent outcomes of moderate to high altitude residence in various countries. Confounding factors, such as variations in the access to clinical facilities or selection biases of investigated populations, may contribute to the equivocation of these observations. We interrogated a dataset of the complete population of an Alpine country in the center of Europe with relatively similar testing and clinical support conditions across altitude-levels of residence (up to around 2000 m) to assess altitude-dependent mortality from COVID-19 throughout 2020. While a reduced all-cause mortality was confirmed for people living higher than 1000 m, no differences in the mortality from COVID-19 between the lowest and the highest altitude regions were observed for the overall population and the population older than 60 years as well. Conversely, COVID-19 mortality seems to have been reduced in the very old (>85 years) women at moderate altitudes.

## 1. Introduction

Residence of populations at high altitude environments (>2500 m) for many generations is associated with substantial adaptations (in particular related to oxygen delivery) to cope with altitude-related hypoxic stress. While adaptations to moderate altitudes, i.e., from 1000 m up to about 2000 m, are less well investigated, it has been repeatedly demonstrated that permanent residence at such altitudes is associated with beneficial health effects and lower mortality rates overall and from various diseases [1,2]. Lower mortality from cardiovascular and cerebrovascular diseases [1,2], from certain cancers [1], i.e., male colorectal and female breast cancer, were reported for people residing at the moderate altitudes of Alpine regions in Europe. Similarly, studies from the Unites States suggest protective effects of moderate-altitude residence on dying from ischemic heart disease [2]. In contrast, living at those moderate altitudes may increase mortality from respiratory diseases [2,3]. Therefore, it could be speculated that mortality from coronavirus disease (COVID-19), an infectious disease caused by the acute respiratory syndrome coronavirus 2 (SARS-CoV-2), are higher in moderate and high-altitude populations. A study on people living at about 3000 m in Peru indeed demonstrated increased inflammation levels [4]. On the other hand, beneficial altitude effects on cardiovascular and metabolic risk factors [5] may counteract those negative effects. However, reports on mortality from COVID-19 in moderate- and high-altitude regions are equivocal and the suggested possible factors mediating such potential effects have been previously critically summarized [6]. Retrospective analysis (January to April 2020) of population-level data from the United States and Mexico reveal higher mortality rates from COVID-19 in people living at or above 2000 m when compared to those living below 1500 m [7]. In contrast, a recent study from Ecuador reported that absolute COVID-19-related excess mortality is lower in areas located at higher altitudes (1500–5500 m) compared to lower ones (<1500 m) [8]. Moreover, high-altitude (Ecuador, 2850 m) was shown to be associated with better short-term survival of critically ill COVID-19 patients [9]. On the one hand, different altitude levels (from moderate to high) may differentially affect COVID-19 outcomes. While at higher altitudes, extreme environmental conditions like hypobaric hypoxia, increased solar radiation, low ambient temperature, and reduced air pollution, are likely major modulators of COVID-19 mortality [8], altered prevalence of cardiovascular risk factors and lifestyle behaviors like physical activity and diet may be more important modulators at moderate altitudes [5]. On the other hand, these ambiguous findings may be explained by potential confounding factors such as differences in the viral load related to population density and/or exposure time to aerosols [10], the availability of services to treat or diagnose COVID-19 [11], which are often less available in higher altitude regions [12]. Such confounders are less relevant in moderate altitude regions of Austria, where access to health care facilities is mostly secured. Therefore, we aimed to assess altitude-dependent mortality data (overall and from COVID-19) of the entire Austrian population in 2020.

## 2. Materials and Methods

Mortality data of the entire Austrian population in 2020, i.e., overall mortality and mortality from COVID-19 (ICD-10: U071) were extracted from the Austrian Causes of Death Statistics (located at Statistics Austria). Data were compared for the altitude sub-categories <500 m, 500–1000 m, and >1000 m (up to the highest community of about 2000 m). Generally, the following altitude categories are widely accepted: low altitude <1500 m, high altitude 1500–3500 m, very high altitude 3500–5500 m and extreme altitude >5500 m [13]. The 1500–3500 m altitude category is often sub-divided into moderate altitude 1500–2500 m and high altitude 2500–3500 m [14]. Accordingly, our data cover low and moderate altitude categories, which are further sub-divided into 3 ranges, similar to our previous study [1].

For each case of death, the respective altitude of the place of residence has been added through data merging via the municipality code. Altitude-dependent mortality rates (MRs, 95% confidence intervals, 95% CI) per 100,000 for the respective sub-population and/or rate ratios are reported for both sexes of the overall population and for the population aged 60+, and depicted for various age groups. We used (crude) age-specific death rates for both sexes. As we performed only comparisons within the same population, no standardization was applied. For the calculation of differences between MRs and rate ratios (95% CI) the calculator from SciStat.com was used: https://www.scistat.com/statisticaltests/ (last accessed on 3 February 2022) which is based on the guidelines of Sahai and Khurshid [15] and Daly (1998) for the calculation of the 95% CI for the MRs [16]. 

## 3. Results

In 2020, the total Austrian population consisted of 4,378,772 male and 4,522,292 female citizens. 41,980 males and 43,145 females died from all causes and 3293 males and 2931 females from COVID-19. 

An altitude map of Austria is presented in Figure 1.

Altitude-dependent numbers of the population for both sexes and respective numbers of deaths from all causes and those from COVID-19 in 2020 are shown in Table 1.

Overall COVID-19 MRs (per 100,000 of population in 2020) were 75.2 for males and 64.8 for females. A comparison of excess mortality between European countries in 2020 (on a monthly basis) can be obtained here: https://ec.europa.eu/eurostat/web/products-eurostat-news/-/ddn-20210216-2 (accessed on 20 November 2022).

More detailed information on the altitude dependent all-cause MRs and COVID-19 MRs for specific age groups of both sexes are shown in Table 2 and Figure 2.

The all-cause MRs (per 100,000 of the respective population) were lower for men and women living above 1000 m compared to those residing at or below 1000 m of altitude (Table 2 and Figure 2). While more males than females died from COVID-19 in all altitude categories (these differences were even clearer for the 60+ categories), there were no differences for mortality from COVID-19 between the lowest and the highest altitude regions for either sex. (Table 2). MR ratios (95% CI) between altitudes >1000 m and <500 m of the population aged 60+ from all causes were 0.84 (0.78–0.91), *p* < 0.001, for males and 0.77 (0.71–0.84), *p* < 0.001, for females, and those for COVID-19 were 1.20 (0.94–1.51), *p* = 0.12, for males and 0.79 (0.57–1.07), *p* = 0.12, for females. 

## 4. Discussion

Altitude-related mortality data from the year 2020 of an Alpine population (Austria) confirm the beneficial effects of living above 1000 m up to about 2000 m on the mortality from all causes, but do not indicate benefits of moderate-altitude (1000–2000 m) residence on the overall mortality from COVID-19 for the total population and the population older than 60 years. However, favorable altitude effects become particularly evident within the highest age groups and the comparably low COVID-19 mortality in very old males (90+) and females (85+) may indicate an altitude effect in this age group (Figure 2). However, due to the very low fatalities in the higher age groups in moderate altitude, caution is needed for comparisons with the respective populations at lower altitudes. 

Only very few comparable data are available on the COVID-19 mortality at moderate altitudes (1000–2000 m) as evaluated in the present study. One study from Saudi Arabia found that people living at higher altitudes (about 1500 m) developed less severe forms of COVID-19 compared to sea-level residents [17]. A negative association between case-fatality ratio for COVID-19 patients and altitude (sea level to 3500 m) was found in a large cohort of Mexican patients. [18]. However, this and other studies pointed out the important impact of comorbidities like obesity, diabetes mellitus and hypertension on the outcome of COVID-19, which may confound reported altitude effects [18,19]. Although Quevedo-Ramirez et al. showed a decrease in the excess mortality with increasing altitude (sea level up to >4000 m) in Peru, lower values were rather typical for altitudes between 1000 and 2000 m [20]. These authors refer to methodological difficulties (official COVID-19-death recordings, geographical variations, etc.) when assessing altitude-associated COVID-19 mortality [20]. They support suggestions of a decreased virulence of SARS-CoV-2 at high altitude, likely associated with the smaller populations and a lower population density at higher altitudes. Further population-level data from the United States and Mexico reveal higher MRs from COVID-19 in people living at or above 2000 m when compared to those living below 1500 m [7]. Noteworthy, among Mexican subjects younger than 65 years, the risk of death was higher in inhabitants of areas above 2000 m versus below 1500 m, but no association was found between altitude and COVID-19 mortality among Mexican women or among Mexican subjects aged 65+ [7]. COVID-19 MRs reported from different countries are difficult to compare due to differences in observation periods and actual infection spreading. To provide some examples, the reported MRs (recorded between 20 January and 13 April 2020) by Woolcott and colleagues in U.S. counties located at or above 2000 m compared to those below 1500 m were 12.3 vs. 3.2 per 100,000 [7]. In contrast, Ortiz-Prado and colleagues found higher MRs (during the first year of the COVID-19 pandemic) at lower altitudes (<2500 m) compared to higher ones (301 vs. 242 per 100,000) [8].

A closer look on potential confounding factors such as environmental factors others than altitude per se, the availability of services to treat, diagnose, or test for COVID-19, assessing the cause of death, etc., which are mostly less available in high-altitude regions, indicate possible fallacies of a “protective” effect of high-altitude residence [12].

Lower temperature and humidity [21] and elevated air pollution [22] may contribute to the increase of mortality from COVID-19. While low temperature and humidity are characteristic for higher elevations, air pollution depends on various factors, e.g., on main traffic routes, industrial emissions, domestic heating, and local wind systems. 

Due to a well-established infrastructure and comparable medical care across the country, access to medical services is similar across the analyzed altitude ranges in Austria, providing a more accurate assessment of the altitude-dependent mortality from COVID-19. Although transportation of COVID-19 patients to hospitals in lower regions cannot be entirely excluded, this is likely of minor importance, since specialized hospitals are closely situated to most villages included in the analyses. It seems that positive effects of living at moderate altitudes on cardiovascular and all-cause mortality [1] that are likely mediated by (sex-specific) differences in the prevalence of cardiovascular risk factors and life-style behavior between low and moderate-altitude residents in Austria [5] did not protect from COVID-19 related mortality In agreement with others [6,23], the assumption that genetic high-altitude adaptations could provide some protection from severe COVID-19, for example lower expression of Angiotensin Converting Enzyme 2 (ACE 2) receptors or variations in the binding affinity of ACE 2 for SARS-CoV-2 is not supported by our findings, at least for the moderate altitudes of the present study. 

The potentially favorable altitude effects on COVID-19 mortality within the highest female age groups warrants verification and further investigation. Such long-lived people, especially when grown-up at altitude, may be exceptionally resistant, with a surprising capacity to recover from illness and complications, associated with an optimal performance of their immune system [24].

## 5. Conclusions

Moderate altitude residence was not associated with altered COVID-19 mortality. Beneficial effects of living at moderate altitude on overall mortality, mortality from cardiovascular diseases and certain cancers do not apply to COVID-19 mortality. It is conceivable that particularly patients suffering from COVID pneumonia may have been negatively affected by residence at higher altitudes. It will be interesting to see whether mortality during the Omicron wave, which is characterized by a lower COVID-19 pneumonia prevalence, would be reduced in people living at moderate altitude.

## Figures and Tables

**Figure 1 ijerph-19-16074-f001:**
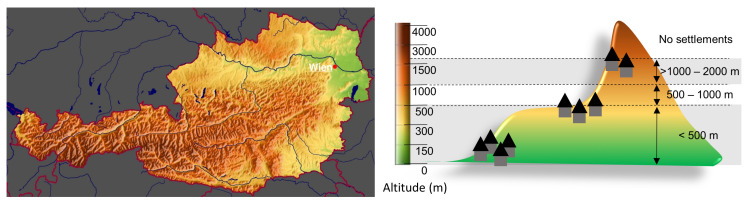
A map depicting altitudes in Austria (**left**) and visualization of the categories of altitude residence used here (**right**); <500 m, 500–1000 m and >1000 m (the highest Austrian villages are at about 2000 m). The map has been obtained from http://www.ginkomaps.com, accessed on 13 March 2022 and modified. Wien (Vienna); capital of Austria.

**Figure 2 ijerph-19-16074-f002:**
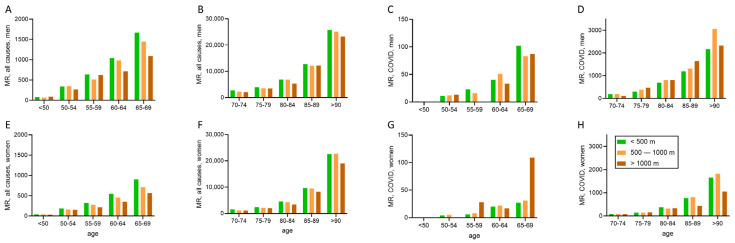
Altitude-dependent mortality rates (MR) of different age-groups of both sexes ((**A**–**D**): men; (**E**–**H**): women) for all causes (**A**,**B**,**E**,**F**) and COVID-19 (**C**,**D**,**G**,**H**). Note the different y-axis, due to lower MRs in lower age-groups.

**Table 1 ijerph-19-16074-t001:** Altitude-dependent numbers (for both sexes) of the overall population and those aged above 60 years, and respective numbers of deaths from all causes and from COVID-19 in 2020.

Altitude	Age Group	Sex	Population	Deaths from All Causes	Deaths from COVID-19
<500 m	Overall	Males	3,163,661	30,092	2186
		Females	3,284,809	31,345	2073
	60+	Males	710,509	26,168	2079
		Females	888,824	29,248	2034
500–1000 m	Overall	Males	1,127,582	11,115	1031
		Females	1,150,578	11,155	813
	60+	Males	280,049	9856	997
		Females	334,817	10,488	798
>1000 m	Overall	Males	87,529	773	76
		Females	86,905	645	45
	60+	Males	21,315	663	75
		Females	23,748	602	43

**Table 2 ijerph-19-16074-t002:** Altitude-dependent mortality rates for both sexes of the overall population and those aged 60+ (per 100,000 of the respective population; and 95% confidence intervals) from all causes and from COVID-19 in 2020.

		Mortality Rate from All Causes		Mortality Rate from COVID-19	
Altitude	Population	Males	Females	Males	Females
<500 m	Total	951	954	69	63
		(941–962)	(944–965)	(66–72)	(60–66) *
	60+	3683	3291	293	229
		(3639–3728)	(3253–3329) *	(280–306)	(219–238) *
500–1000 m	Total	986	970	91	71
		(968–1004) ^§^	(952–988)	(86–97) ^§^	(66–76) *^§^
	60+	3519	3132	356	238
		(3450–3590) ^§^	(3073–3193) *^§^	(334–379) ^§^	(222–256) *
>1000 m	Total	883	742	87	52
		(822–948) ^§&^	(686–802) *^§&^	(68–109)	(38–69) *^&^
	60+	3110	2535	352	181
		(2878–3357) ^§&^	(2336–2746) *^§&^	(277–441)	(131–244) *

* denotes significantly different (*p* < 0.05) between sexes; ^§^ denotes significantly different to altitude <500 m and ^&^ significantly different to 500–1000 m; within the respective mortality causes and age groups.

## Data Availability

Data are available at https://www.statistik.at/ (accessed on 20 November 2022).

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
