# Peer review of "Health Benefits of Residence at Moderate Altitude Do Not Reduce COVID-19 Mortality"

_ijerph, 2022, doi:10.3390/ijerph192316074_

Round 1

Reviewer 1 Report

Nice work on this novel and timely paper which is well prepared. I recommend the following minor edits:

There are working definitions for altitude (low, moderate, high, very high, extreme). I recommend making these changes throughout, citing an authoritative source, and provide the reader with some understanding about the working definitions used within. Moreover, there are ranges used in the current manuscript (e.g., 1500-5000m) that would span several echelon of altitude, a matter that is a simple edit to include the range of altitude cut points. This point is separate from the 500m cut points used for the current analysis. That is, just indicate how your cut points relate to common working definitions of altitude.

Other potentially confounding factors need to be addressed for the reader. For instance, you haven't accounted for the number of people transported to lower elevation to treat their covid case, for those that are provided supplemental O2 (effectively resolving the lower partial pressure of O2 at a given altitude), the fact that few instances of clinically significant cases of altitude sickness are observed at the upper altitudes examined in the current study. All of these are editorial fixes, but should be addressed with appropriate referencing.

Statistical denotations are needed in Fig 2

Statistical denotations are needed in Tables 1,2

It seems like there is an inference to populations normalized analyses between and within a given altitude (with and without sex considerations), but I do not see these data. Please provide these analyses for the reader. 

line 133 needs editing

Author Response

REVIEWER 1:

Nice work on this novel and timely paper which is well prepared. I recommend the following minor edits:

There are working definitions for altitude (low, moderate, high, very high, extreme). I recommend making these changes throughout, citing an authoritative source, and provide the reader with some understanding about the working definitions used within. Moreover, there are ranges used in the current manuscript (e.g., 1500-5000m) that would span several echelon of altitude, a matter that is a simple edit to include the range of altitude cut points. This point is separate from the 500m cut points used for the current analysis. That is, just indicate how your cut points relate to common working definitions of altitude.

Reply: We thank the reviewer for the favorable opinion on our manuscript and the really constructive feedback that helped us to further improve the article.

We now describe the currently used systems of altitude ranges. We added to the first paragraph of the method section:

Generally, the following altitude categories are widely accepted: low altitude < 1,500 m, high altitude 1,500 3,500 m, very high altitude 3,500-5,500 m and extreme altitude > 5,500 m [13]. The 1,500 – 3,500 m altitude category is often sub-divided into moderate altitude 1,500 – 2,500 m and high altitude 2,500 – 3,500 m [14]. Accordingly, our data cover low and moderate altitude categories, which are further sub-divided into 3 ranges, similar to our previous study [1].

Other potentially confounding factors need to be addressed for the reader. For instance, you haven't accounted for the number of people transported to lower elevation to treat their covid case, for those that are provided supplemental O2 (effectively resolving the lower partial pressure of O2 at a given altitude), the fact that few instances of clinically significant cases of altitude sickness are observed at the upper altitudes examined in the current study. All of these are editorial fixes, but should be addressed with appropriate referencing.

Reply: This is a good point. We already stated: “Due to a well-established infrastructure and comparable medical care across the country, access to medical services is similar across the analyzed altitude ranges in Austria, providing a more accurate assessment of the altitude-dependent mortality from COVID-19”. Now we added the following statement:

Although transportation of COVID-19 patients to hospitals in lower regions cannot be entirely excluded, this is likely of minor importance, since specialized hospitals are closely situated to most villages included in the analyses.

Statistical denotations are needed in Fig 2

Statistical denotations are needed in Tables 1,2

Reply: Data presented in table 1 refer to the overall Austrian population and were further used for mortality calculations presented in table 2. Beside 95% confidence intervals, also statistically significant differences (p < 0.05) between sexes and altitude regions are provided in this table. Figure 2 was intended to compare descriptively (graphically) patterns of altitude-dependent mortality rates (from all causes and COVID-19) of different age-groups of both sexes. Due to the small numbers of fatalities in the higher age classes at moderate altitudes statistical denotations appear inappropriate here; we do, however, indicate this limitation in the discussion now:

However, due to the very low fatalities in the higher age groups in moderate altitude, caution is needed for comparisons with the respective populations at lower altitudes.

It seems like there is an inference to populations normalized analyses between and within a given altitude (with and without sex considerations), but I do not see these data. Please provide these analyses for the reader. 

Reply: We used (crude) age-specific death rates for both sexes (overall and 60+ in table 2, various age-groups in figure 2). As we performed only comparisons within the same population, no standardization was applied. This has now been mentioned at the end of the method section.

line 133 needs editing

Reply: We thank the reviewer for pointing us to this previously complicated sentence. We changed to:

“A negative association between case-fatality ratio for COVID-19 patients and altitude (sea level to 3,500 m) was found in a large cohort of Mexican patients.”

Reviewer 2 Report

The objective of the study "Health benefits of residence at moderate altitude do not reduce COVID-19 mortality " was to assess altitude-dependent mortality data (overall and from COVID-19) of the entire Austrian population in 2020. This study is interesting and useful for assessment of the health consequences of living at high altitude in a country with well-established infrastructure and comparable medical care across country.The paper is well written.However, I have the following comments:

Line 35: Please provide a reference for thresholds of moderate and low altitudes. Since some guide lines classify moderate altitude over 1500 meters above sea level.

Line 74: Why was this classification used? could explain it.

Line 135: Why did you not include these comorbidities in your analysis? , it would be interesting to include it in order to provide more support to the discussion of your results.

Line 168: Regarding, by assuming that negative consequences of altitude on the mortality from respiratory diseases are partly compensated by positive effects of living at moderate altitudes on cardiovascular and all-cause mortality. I think that this statement cannot be made, the morbidity and mortality indicators correspond to the same period, to the same austrian population density? Please modify it.

Line 161. Is it possible to associate these environmental factors with genetic characteristics of high altitude inhabitants such as the alteration of the Renin Angiotensin System (SRA)?

Line 186. I think that it cannot be concluded regarding all respiratory diseases, because the pathophysiological phenomenon is not explained, it should be specifically indicated that they are referring to covid pneumonia, or directly COVID.

Line at high altitude in a country with well-established infrastructure and comparable medical care across country.The paper is well written.However, I have the following comments:ent mortality data (overall and from COVID-19) of the entire Austrian population in 2020. This study is interesting and useful for assessment of the health consequences of living at high altitude in a country with well-established infrastructure and comparable medical care across country.The paper is well written.However, I have the following comments:

Author Response

REVIEWER 2:

The objective of the study "Health benefits of residence at moderate altitude do not reduce COVID-19 mortality " was to assess altitude-dependent mortality data (overall and from COVID-19) of the entire Austrian population in 2020. This study is interesting and useful for assessment of the health consequences of living at high altitude in a country with well-established infrastructure and comparable medical care across country. The paper is well written. However, I have the following comments:

Reply: We are very grateful for the favorable evaluation of our manuscript and the really constructive and helpful feedback. Please find below our replies to all comments and indications on the according changes performed in the manuscript.

Line 35: Please provide a reference for thresholds of moderate and low altitudes. Since some guide lines classify moderate altitude over 1500 meters above sea level.

(please see next point)

Line 74: Why was this classification used? could explain it.

Reply: Thank you for this point that was also addressed by another reviewer. We now added to the method section:

“Generally, the following altitude categories are widely accepted: low altitude < 1,500 m, high altitude 1,500 – 3,500 m, very high altitude 3,500-5,500 m and extreme altitude > 5,500 m [13]. The 1,500 – 3,500 m altitude category is often sub-divided into moderate altitude 1,500 – 2,500 m and high altitude 2,500 – 3,500 m [14]. Accordingly, our data cover low and moderate altitude categories, which are further sub-divided into 3 ranges, similar to our previous study [1].“

Line 135: Why did you not include these comorbidities in your analysis? , it would be interesting to include it in order to provide more support to the discussion of your results.

Reply: We agree that this would be highly interesting information. Unfortunately, we only have information on death rates but not on associated comorbidities.

Line 168: Regarding, by assuming that negative consequences of altitude on the mortality from respiratory diseases are partly compensated by positive effects of living at moderate altitudes on cardiovascular and all-cause mortality. I think that this statement cannot be made, the morbidity and mortality indicators correspond to the same period, to the same austrian population density? Please modify it.

Reply: Thank you, we agree. We modified as follows:

“It seems that positive effects of living at moderate altitudes on cardiovascular and all-cause mortality, that are likely mediated by (sex-specific) differences in the prevalence of cardiovascular risk factors and life-style behavior between low and moderate-altitude residents in Austria, did not protect from COVID-19 related mortality.”

Line 161. Is it possible to associate these environmental factors with genetic characteristics of high altitude inhabitants such as the alteration of the Renin Angiotensin System (SRA)?

Reply: This is an interesting suggestion. We don’t think that such genetic characteristics are important at the moderate altitudes of the present study but cannot exclude it based on the available data., We therefore included the following statement:

“In agreement with others [6], the assumption that genetic high-altitude adaptations could provide some protection from severe COVID-19, for example lower expression of Angiotensin Converting Enzyme 2 (ACE 2) receptors or variations in the binding affinity of ACE 2 for SARS-CoV-2, is not supported by our findings, at least for the moderate altitudes of the present study.” 

Line 186. I think that it cannot be concluded regarding all respiratory diseases, because the pathophysiological phenomenon is not explained, it should be specifically indicated that they are referring to covid pneumonia, or directly COVID.

Reply: Yes, we agree and modified this part as follows:

“Beneficial effects of living at moderate altitude on overall mortality, mortality from cardiovascular diseases and certain cancers do not apply to COVID-19 mortality. It is conceivable that particularly patients suffering from COVID pneumonia may have been negatively affected by residence at higher altitudes.”

And added:

“It will be interesting to see whether mortality during the Omicron wave, which is characterized by a lower COVID-19 pneumonia prevalence, would be reduced in people living at moderate altitude.”